# Hemispherical Pediatric High-Grade Glioma: Molecular Basis and Therapeutic Opportunities

**DOI:** 10.3390/ijms21249654

**Published:** 2020-12-17

**Authors:** Santiago Haase, Fernando M. Nuñez, Jessica C. Gauss, Sarah Thompson, Emily Brumley, Pedro Lowenstein, Maria G. Castro

**Affiliations:** 1Department of Neurosurgery, University of Michigan Medical School, Ann Arbor, MI 48109, USA; sthaase@umich.edu (S.H.); fmnunez@umich.edu (F.M.N.); jcwilli@umich.edu (J.C.G.); thompsar@umich.edu (S.T.); ebrumley@umich.edu (E.B.); pedrol@umich.edu (P.L.); 2Department of Cell and Developmental Biology, University of Michigan Medical School, Ann Arbor, MI 48109, USA

**Keywords:** pHGG, HGG, pediatric high-grade glioma, hemispheric pHGG, G34R, ATRX, cancer, epigenetics

## Abstract

In this review, we discuss the molecular characteristics, development, evolution, and therapeutic perspectives for pediatric high-grade glioma (pHGG) arising in cerebral hemispheres. Recently, the understanding of biology of pHGG experienced a revolution with discoveries arising from genomic and epigenomic high-throughput profiling techniques. These findings led to identification of prevalent molecular alterations in pHGG and revealed a strong connection between epigenetic dysregulation and pHGG development. Although we are only beginning to unravel the molecular biology underlying pHGG, there is a desperate need to develop therapies that would improve the outcome of pHGG patients, as current therapies do not elicit significant improvement in median survival for this patient population. We explore the molecular and cell biology and clinical state-of-the-art of pediatric high-grade gliomas (pHGGs) arising in cerebral hemispheres. We discuss the role of driving mutations, with a special consideration of the role of epigenetic-disrupting mutations. We will also discuss the possibilities of targeting unique molecular vulnerabilities of hemispherical pHGG to design innovative tailored therapies.

## 1. Introduction

Pediatric high-grade gliomas (pHGG) are highly invasive brain tumors accounting for approximately 15% of all central nervous system tumors in children and adolescents [1]. The World Health Organization (WHO) classifies non-brainstem pHGGs as anaplastic astrocytoma (WHO grade III) and glioblastoma (GBM; WHO grade IV), reflecting their aggressive nature and resistance to conventional treatment [2]. 

The Central Brain Tumor Registry of the United States (CBTRUS) reports that anaplastic astrocytoma (AA) is most prevalent in children ages 5–9 years and that glioblastoma (GBM) is most prevalent in children ages 10–14 years [3]. The histological characteristics of pediatric high-grade gliomas include hypercellularity, nuclear atypia, abnormally high mitotic activity, and increased angiogenesis and/or necrosis, the latter two associated primarily with GBM morphology [4]. Patients with pHGG exhibit an array of symptoms consistent with CNS malignancies, such as focal neurological deficits and cranial nerve palsies, with individual presentation largely dependent on the patients’ age and the location of the tumor [5]. However, due to their proliferative nature, high-grade gliomas have shorter durations between symptom onset and diagnosis compared to tumors of lower grade, precluding the clinical advantages of early detection [6,7]. High-grade glioma comprises 8 to 12% of all central nervous system (CNS) pediatric tumors and have an incidence of approximately 0.85 per 100,000 children [3]. One third of pHGG are supratentorial, and among these, half of them are hemispherical pHGG [8]. Thus, cortical pHGG incidence is approximately 0.12 per 100,000 children, affecting mainly adolescents aged 15–19 years [8,9].

The prognosis for pHGG is dismal, with an overall median survival of 9-15 months and a 5-year survival rate of less than 20% [10]. Surgical intervention of cortical pHGG patients includes tumor resection and biopsy. Total tumor resection is often impossible in pHGG, as these infiltrative tumors often progress into normal tissue beyond surgical margins [11]. Furthermore, the extent of resection (EOR) is often limited in order to preserve the neurological functions of delicate brain regions surrounding the tumor. Nevertheless, EOR is a significant prognostic marker for overall survival in pediatric patients with malignant hemispheric gliomas [12]. Although surgery is the primary intervention for treatment of non-brainstem pHGGs, it is not curative. The standard of care also includes radiation therapy for pHGG patients above three years of age, typically 50–60 Gy delivered over 3–6 weeks [6]. Currently, no chemotherapeutic treatments are involved in the standard therapy for pHGG; however, various treatments are being tested in clinical trials [13]. Despite immense efforts, there are no effective treatment options and pediatric high-grade glioma has become the leading cause of cancer-related death in children and adolescents under 19 [3,4].

## 2. Genetic Alterations on NBS pHGG

Recent advancements in molecular profiling have vastly improved our understanding of pediatric high-grade glioma and have identified unique genetic and epigenetic features of pHGG which had been previously conflated with adult gliomas. Several pathways and molecular alterations were identified in hemispherical pHGG, including the PI3K/AKT, Ras-Raf-MEK-ERK, RB, and p53 pathways [14,15,16,17,18] (Figure 1). Most notably, the discovery of recurrent mutations in the genes encoding histone variants H3.3 (H3F3A) and H3.1 (HIST1H3B/C) demonstrated the unique biology of pediatric brain tumors [10,19,20] (Figure 2). 

Three somatic mutations were identified, encoding amino acid substitutions resulting in replacement of lysine with methionine at residue 27 of histones H3.1 and H3.3 (K27M) in brainstem/midline pHGG or in glycine to arginine or valine substitution at residue 34 (G34R/V) of histone H3.3 in hemispheric pHGG. These mutations rewire the epigenome, resulting in global hypomethylation and disrupting critical regulatory sites of posttranslational modifications [20]. Furthermore, these mutually exclusive mutations are found in gliomas arising in specific anatomical locations and are associated with distinct age groups and survival outcomes [19,21].

### 2.1. Histone Mutations in Hemispheric pHGG

The nucleosome is the elemental unit of the chromatin. It consists of a histone polyprotein octamer with two copies each of histones H3, H4, H2A, and H2B, wrapped almost twice by 147 base pairs (bp) of DNA [22]. The location and density of the nucleosomes within the genome sterically determines the ability of the molecular machinery to access the genome, providing a mechanism of control for genomic processes such as transcription and DNA metabolic processes such as replication, repair, and recombination. Based on this precept, an elegant system regulating chromatin metabolism has evolved [22,23]. This system consists of a code based on histone posttranslational modifications (PTMs) occurring mainly in histone tails, which are histone fragments that protrude from the nucleosome core. For decades, it was thought that the cause of cancer relied on mutations of key genes. It is now clear that aberrant epigenetic changes play a critical role in almost every step of tumor development and progression [24], contributing to tumor heterogeneity and evolution [25], particularly in pediatric cancer [26]. 

Mutations in proteins involved in epigenetic regulation were frequently identified in a wide range of cancers [27]. In particular, mutations in genes related to chromatin metabolism are found more frequently in pediatric cancer, which highlights the importance of epigenetic disruption in oncogenesis [28,29] (Figure 2). Interestingly, epigenetic disruption is also associated with several neurodevelopmental diseases [30]. In 2012, genome sequencing revealed that histones themselves are mutated in cancer. Two independent studies found the frequent occurrence of mutations in histone H3 in pHGG [19,20]. Shortly after, mutations in H3 were found in other cancers arising in childhood to early adulthood such as chondroblastoma [31,32], giant cell tumors of bone [31], chondrosarcoma [31], pediatric soft tissue sarcoma [32], head and neck squamous cell carcinoma [33], acute myeloid leukemia (AML) [34], ependymomas [35], and gangliogliomas [36]. 

Humans encode for different H3 variants. H3.1 and H3.2 are expressed only during the S phase in such a way that they are incorporated into nascent chromatin [37]. H3.1 and H3.2 differ only by 1 amino acid. On the other hand, the replacement variant H3.3 is constitutively expressed and is the substrate for replication-independent nucleosome assembly [38]. H3.3 replaces replication-dependent H3 at active genes as part of an epigenetic regulatory process, and this process has consequences on gene expression and other chromatin mechanisms, such as DNA repair [39,40]. The differences on the PTMs of the replacing H3.3 might pose consequences to the local properties of chromatin [41]. Additionally, H3.3 differs from H3.1 and H3.2 in five amino acid positions. These substitutions are mainly at the core of the histone protein and are responsive to the binding specificity of the chaperone proteins, which control the differential deposition of the nucleosomes and determine the differential localization patterns of the H3 variants within the genome [42,43,44,45]. H3.3 is encoded by 2 genes: H3F3A and H3F3B. H3.1 has 10 different alleles, and H3.2 has 3. Other H3 histones are expressed in a tissue specific manner (H3.5, H3.X, and H3.Y) [46], and CENP-A is a histone of the H3 family that is deposited only at centromeres [47]. Two specific complexes that deposit H3.3 were identified: HIRA (histone cell cycle regulator) deposits H3.3 at transcriptionally active euchromatic regions when they become transiently depleted of nucleosomes [48,49,50]; additionally, H3.3 is deposited into heterochromatic regions, including telomeric and pericentromeric regions, by the DAXX (death-domain associated protein)/ATRX (thalassemia/mental retardation syndrome X-linked) complex [51]. As H3.1 and H3.2 are incorporated into nucleosomes in a strictly replication-dependent manner, the proportion of these H3 in chromatin can vary widely among different cells according to the replicative rate, and in non-actively dividing cells, such as neurons, H3.3 will account for most of the total H3 in chromatin.

#### 2.1.1. G34 Mutations in Cortical pHGG

Mutations on H3.3 at G34 were identified in approximately 20% of the pHGG located on the cerebral hemispheres [10]. In these patients, only one copy of the H3F3A allele is mutated. While mutations at lysine 27 (K27M) in H3.1 and H3.3 occurring in midline gliomas and mutations at lysine 36 (K36M) found in most chondroblastoma and other childhood cancer are directed to positions that are normally subjected to posttranslational modifications, the H3.3 glycine at amino acid 34 is not posttranslationally modified, although G34 resides on the histone tail at 2 amino acids of distance from the lysine 36 position.

Tumors harboring the H3.3 G34R/V mutations are confined to the cerebral hemispheres, most commonly to the parietal and temporal lobes [18,19,21,52,53]. Patients with G34R/V mutant tumors represent an older cohort of pHGG, with a median age of 15 years and overall median overall survival of 18 months, with 2-year survival of 27.3% [10]. The G34R/V mutation defines a molecular subgroup of pHGG associated with loss of function mutations in tumor suppressor protein 53 (TP53) and mutations in ATRX or DAXX [10,19,54]. ATRX and DAXX are components of a chromatin remodeling complex necessary for incorporation of histone H3.3 at the pericentric heterochromatin of telomeres [18]. The loss of function of ATRX and, less frequently, DAXX impairs telomeric stability and results in telomerase-independent maintenance through alternative lengthening of telomeres (ALT) in H3.3G34R/V mutant tumor cells [10]. The ALT phenotype of G34R/V is strongly correlated to the loss of TP53/ATRX/DAXX; however, the role of TP53 in this process is not yet clear [18,19]. These tumors also show greater genetic instability and enriched chromosomal copy-number alterations (CNAs) [19]. Although the G34R/V mutant residue is not posttranslationally modified itself, the mutant histone alters epigenetic regulation of the lysine residue at position 36 (H3F3A-K36). The K36 position can be methylated (K36me1/2/3) or acetylated (K36Ac), and trimethylation (me3) is a mark of transcriptional activation and is also implicated in alternative splicing and DNA repair mechanisms [18,55]. H3.3 G34R/V mutant tumors display a global DNA hypomethylation phenotype but have differential distribution of H3K36me3 with distinct gene targets involved in cortical development and stemness [18]. Notably, oncogene MYCN was found to be significantly upregulated by increased distribution of H3K36me3 in H3.3 G34R/V tumors [10,56]. DNA methylation profiling has also revealed that H3.3 G34R/V is the only histone mutant subgroup of pHGG to display significantly elevated methylation levels at the promoter of O-6-Methylguanine-DNA Methyltransferase (MGMT), a DNA repair enzyme [10,57].

#### 2.1.2. Molecular Mechanisms of G34 Mutations

An interesting challenge is to elucidate how G34 mutations on the H3 histone exert dominant effects, given that 30 alleles encoding H3 isoforms exist. The first molecular studies revealed that K36 trimethylation is affected when a mutant G34 is expressed. Nonetheless, K36 trimethylation is only decreased on the G34 mutant histone protein itself, with no consequences on K36me3 distribution at the wild type histone. This mechanism does not fully explain the dominant effect of G34 mutations on pHGG. Mammalian cells contain at least eight H3K36 methyltransferases, and among these, NSD1–3 and SET Domain Containing 2 protein (SETD2) are considered major H3K36 methyltransferases. Only SETD2 can catalyze H3K36 trimethylation, whereas the rest of the enzymes perform H3K36 mono- and/or di-methylation. On the other hand, lysine 36 acetylation is carried out by Gcn5 [58]. Although K36Ac was associated with active transcription, the specific functions of this mark in human cells are poorly understood, and the effects of G34 mutations on the deposition of this mark have not been studied.

Several studies demonstrated that H3.3 G34 mutations can inhibit the interaction of tri-methylase SETD2 [59,60,61]. Structural studies demonstrated that SETD2 performs its activity by binding a tract of the histone tail to a narrow channel of the protein. SETD2 fails to bind to H3.3 G4R/V because the channel cannot accommodate any amino acid larger than glycine. This would explain why the G34 mutated histone itself has lower levels of K36me3. Although this channel is well conserved among K36 methylases [59,60], mono and di-methylases are thought to have structural differences that would allow for accommodation of the H3G34R/V mutations, which explains why K36me1/2 are not affected by this mutation [62]. 

One of the consequences of the expression of H3.3G34R/V is the gain in H3K36me3 at specific loci [56]. A recent study suggests that the increase in K36me3 is due to inhibition of KDM4 demethylases by the mutant histone [63]. H3.3G34R sequesters KDM4, inhibiting its activity, and consequently, the expression of G34R causes an abnormal accumulation of K36me3 and of H3K9me3, marks that are normally removed by KDM4 [63]. This mechanism would be consistent with a dominant role for G34 mutations in pHGG. This model, firstly, predicts a reduction of the global KDM4 activity and, secondly, may imply that the local downregulation of K36me3 can be a consequence of the inability of SETD2 to interact with the mutant histone that is strongly bound to KDM4.

H3K36 methylation has been shown to antagonize with H3K27me3. This phenomenon is associated with the inhibition of interaction of the K27 trimethylating complex PCR2 with H3 when the histone is marked with K36 trimethylation [64,65]. In line with this, it was shown that chondroblastoma cells harboring K36M mutations exhibit a global increase in K27me3 [32]. This phenomenon was verified by expressing an exogenous H3.3K27M in cell culture models. The results suggest that a global loss of K36me2/3 caused by the K36M mutation allows for spread of K27me3 by means of the activity of PCR2 in nucleosomes containing H3 K36me1 or non-methylated in K36. It is possible that expression of the H3.3G34 mutations have local effects on the deposition of the H3K27me3 mark.

H3.3G34R mutations not only affect the distribution of histone marks but also were demonstrated to interfere with the interaction of the histone tail with proteins that recognize K36 marks. For example, the mismatch repair (MMR) protein MutSα, which in normal conditions recognizes specifically K36me3 in replicating chromatin to prime mismatch repair, is inhibited by G34 mutations [61]. This has consequences on genomic instability and mutation rate. More recently, a specific reader for H3.3K36me3 was identified. This reader, ZMYND11, is involved with Pol II elongation and has a role in splicing. Structural in vitro analysis suggests that the interaction of ZMYND11 is impeded by G34 mutation [66]. Some of the questions that arose when these mutations were described were why the mutations only occur on the H3F3A allele and not on the H3F3B, why the mutation is exclusive to H3.3, and why R mutations are prevalent over V. Another intriguing finding was that pHGG harboring G34 mutations frequently have mutations in ATRX or DAXX, proteins that are part of the DAXX/ATRX complex, involved in depositing H3.3 into heterochromatin. The nature of the interaction between H3.3 and ATRX/DAXX mutations in driving pediatric gliomas remains to be elucidated [51].

#### 2.1.3. G34 Mutations and DNA Repair

The G34 mutations were associated with defects on DNA repair responses (DDR). The consequences of H3.3-G34R expression were studied in yeast. In contrast to mammal cells, only a single protein, SET2, performs all the methylation PTMs on H3K36 in yeast. In this context, it was observed that H3F3A-G34R expression leads to an accumulation of K36me2 and a reduction of K36me3 and K36Ac levels. This suggests that the mutation affects only the trimethylation reaction performed by SET2, but it does not affect the mono-methylation and di-methylation reactions carried out by the same enzyme. G34R expression was also correlated with genomic instability, as evidenced by an increase in rearrangements and chromosome mis-segregation. G34R-mutated yeast were also more sensitive to compounds that affect DNA replication or induce DNA damage on DNA-replicating cells but not general DNA damage. The authors demonstrated that these results are linked to a defect of G34R mutant yeast to repair via the homologous recombination (HR) pathway. Remarkably, these defects were not observed in SET2 defective yeast, indicating that the decrease of K36me3 is not sufficient to account for the effects of G34R expression in yeast. Although these results link G34R mutations with DNA repair and replication defects, it should be considered that the epigenetic regulation mechanisms between yeast and mammals are strongly different. Noteworthily, the differences on DNA repair responses between G34R mutant and SETD2-deffective non-brainstem (NBS) pHGGs indicate that each oncogenic lesion impacts other epigenetic mechanisms apart from the evident effect on H3K36me3.

One study provides a link between G34/SETD2 mutations and DNA repair [67]. It describes that K36me3 is required to recruit hMutSα, one of the protein complexes that recognize mispairs when replicating DNA to induce mismatch repair (MMR). As a consequence, SETD2-defective cells, which have lower H3K36me3 levels, display microsatellite instability (MSI) and increased mutation rate. In this sense, mutations affecting H3K36me3 display a MMR deficient-like phenotype, a phenotype frequently described in many cancer types [68]. Studies in leukemia models, where SETD2 mutations occur, indicate that these mutations ablate the cells’ ability to recognize DNA damage, and as a consequence, DNA damage-induced apoptosis is blocked in these cells [69]. This makes SETD2 knockout cells more resistant to DNA damage. Another study confirms that G34R/V mutations themselves have a similar effect on MMR pHGG cells [61]. This work demonstrates that G34R/V-mutated H3F3A interacts deficiently with the hMutSα complex, likely as a consequence of a decrease in K36me3 in the histone. pHGG cells genetically engineered to express ectopically H3F3A-G34R exhibited increased mutation compared with isogenic H3F3A controls and the same trend on pHGG patient sample data.

Other studies further associate G34 mutations with DNA repair mechanisms. A work investigated the role of H3.3 in DNA repair, finding that the histone variant is necessary for proper repair of UV damage and DNA repair in DNA replication forks [39]. Noteworthily, G34 mutant cells exhibited increased UV sensitivity. Additionally, H3K36 marks not only are required for a functional DDR but also were demonstrated to coordinate DNA repair pathway choice (HR or NHEJ) after double-strand break damage [70]. Of note, the promoter of the MGMT (O-6-Methylguanine-DNA Methyltransferase) gene, which codes for an enzyme involved in DNA repair, is significantly increased in G34R tumors when compared with other hemispheric pHGGs [10,71], which may result in decreased MGMT expression and increased sensitivity to temozolomide, a DNA alkylating agent. ATRX mutations, which normally cooccur with G34 mutations and SETD2 inactivation, were associated with genomic instability and impaired DNA damaged responses [17]. It remains unclear why multiple mutations affecting DNA repair occur in concert on pHGG.

### 2.2. The Role of ATRX, a Chromatin Regulator

Inactivating mutations in the chromatin remodeler ATRX are found in hemispheric pHGG, showing close association with H3F3A-G34R/V mutations and being present within the IDH and the histone Wt epigenetic subtypes. It was described that ATRX inactivation is necessary to elicit alternative lengthening of the telomere (ALT) mechanism. This mechanism allows the pHGG cells to extend their telomeres without the requirement of telomerase reverse transcriptase (TERT) expression and represents a way to avoid death by catastrophic telomere loss, allowing cancer cells to divide indefinitely and thus enabling cancer progression [51].

The main role of ATRX is related to its functions as a part of the DAXX/ATRX complex, an H3.3 writer that deposits the histone variant mainly into heterochromatin. However, the functions of ATRX exceed this role, and this protein was shown to be involved in the deposition of H3.3 into actively transcribed loci, where it plays a role as an epigenetic regulator [51]. A recent study provides a connection between the role of ATRX as an epigenetic regulator and glioma development [72]. This work demonstrates that ATRX inactivation causes profound changes in cellular morphology and gene expression in mouse neural precursor cells (NPCs). In Wt NPCs, ATRX in transcriptionally active loci and loss of ATRX were associated with expression changes at specific genes. These alterations were further linked with a differentiation of the ATRX^-/-^ NPCs towards an astrocytic lineage, which correlates with an astrocytic histology of ATRX-null pHGG. ATRX loss also was associated with an increased expression of mobility-related genes, resulting in increased migration. Altogether, these results indicate that ATRX may play a role as an oncogene by inducing epigenetic changes on specific genes, targeting differentiation and cell mobility, and the results suggest that these changes are cell context-specific. This may explain why ATRX can drive tumors only in certain cancer types.

Noteworthily, ATRX is also implicated in sustaining chromatin stability and in DNA repair processes. For this reason, ATRX inactivating mutations could provide preneoplastic cells with the ability to acquire additional mutations with a faster rate and to adapt to changing conditions. A more detailed discussion about these mechanisms can be found in [51,73,74]. 

### 2.3. The P53 Pathway in Hemispheric pHGG

The TP53 pathway is the most commonly deregulated pathway in pediatric HGG. Mutations or rearrangements affecting this pathway, which encompass TP53 mutations, CDKN2A (ARF) mutations/rearrangements, and MDM2/4 mutations, are found in about 50% of non-brainstem pHGG cases [52].

TP53 is one of the most commonly mutated genes in human cancer [75] and an important tumor driver in pHGG, occurring in 38% of hemispheric pHGG [76]. Wild type p53 is a tumor-suppressing transcription factor that notably promotes apoptosis and cell cycle arrest and participates in DNA damage response pathways [75]. These functions are disrupted in loss-of-function mutants, which act as key drivers of gliomagenesis [76]. In vitro GBM cells with inactivated-mutant p53 were less susceptible to inhibitors targeting DNA repair [77], highlighting the mutant phenotype’s ability to accumulate mutations at a faster rate and to promote tumor growth.

P53 activity is controlled by MDM2 and MDM4. Under normal conditions, MDM2 and MDM4 mark P53 for degradation, keeping P53 activity low, while p53/MDM2 interaction is disrupted in the presence of stress signals, such as DNA damage or oxidative stress. MDM2 and MDM4 function by downregulating p53 in wild-type cells. These genes are often upregulated in GBM, promoting p53 degradation and inhibition [78]. Mutant P53 also often acts cooperatively with loss of PTEN (Phosphatase and tensin homolog), significantly increasing proliferation speed in double-mutant murine cell lines [79]. On the other hand, the ARF gene, encoded in the CDKN2A locus, regulates the p53 pathway by promoting MDM2 degradation [80]. As a consequence, CDKN2A loss leads to an increased median life for MDM2, which in turn conducts over degradation of P53.

In normal cells, p53 activation induces cell-cycle arrest and apoptosis, regulating cell fate decisions. P53 is activated by posttranslational modifications induced, among other, by severe DNA damage. Once activated, p53 induces p21/WAF1 transcription, which binds to cyclins E/Cdk2 D/Cdk4, and the complexes cause G1 cell cycle arrest. P21-mediated inhibition of these cyclins also prevents pRb phosphorylation, promoting pRb binding to the E2F1 transcription factor. E2F1 silences the transcription of genes that are critical for DNA replication and cell cycle progression. P53 activation also induces G2/M arrest. Checkpoint P53-mediated enforcement gives the cell time to repair DNA damage that otherwise could be catastrophic. P53 also participates in controlling DNA replication, ensuring that it is not replicated a second time and preventing polyploidy events.

Chronic activation of P53 in abnormal contexts, such as when a cell undergoes abnormal division with telomere erosion, may result in senescence, that is, sustained cell-cycle arrest. This process requires p53-mediated p21 activation. The senescent state is stabilized by epigenetic silencing of E2F1 target genes as well as expression of cytokines. In some contexts, P53 can also activate genes that are associated with apoptosis induction. P53 targeting to the mitochondria alone is able to induce mitochondrial outer membrane permeabilization (MOMP), which is a critical step in the intrinsic apoptosis pathway. P53 entering into the mitochondria can also affect inner membrane potential, causing a halt in ATP production and resulting in necrosis.

The major role of P53 in mediating response to DNA damage indicates the critical function of this protein as a tumor suppressor. P53 knockout mice are predictably prone to developing tumors. Preneoplastic and cancer cells adapted to living with unphysiological levels of genomic instability, oxidative stress, and P53 loss impeding cell-cycle arrest, apoptosis, or senescence induction in these conditions, all potent antitumor mechanisms. P53 loss as a tumor initiating event also may allow an environment to become permissive to oxidative stress and DNA damage that would lead quicker to the accumulation of oncogenic mutations.

Besides the evident effect of P53 as a tumor suppressor, P53 expression is rarely lost in hemispheric pHGG. Most of the mutations observed in hemispheric pHGG affect the transcription factor activity of P53 while retaining other functions of the protein. Moreover, gain of function (GOF) mutations were identified in cancer [81,82]. This effect was first evidenced by the fact that the expression of mutated-P53 increased tumorigenic potential. GOF p53 mutations have been associated with enhanced tumor aggressiveness hallmarks, such as proliferation, invasion, and resistance to chemotherapy. Recent research describes that some P53-dominant GOF induces the formation of oligomer aggregates between mutated and Wt TP53, exerting a prion-like dominant inactivating effect over the Wt p53 protein. Other proteins of the p53 family, which have been shown to take the function of P53 when lost, have also shown to be inactivated by this GOF TP53 aggregation mechanism. Another mechanism of GOF mutations is through aberrant transcriptional activity, by means of interactions with distinct transcription factors for which activity leads to tumor initiation and progression. The role of GOF P53 mutations in pHGG is currently understudied, and it is clear that new therapeutic opportunities will arise when we gain better insight into the consequences of mutant P53 expression in the pHGG context.

### 2.4. PDGFR Mutations

Platelet-derived growth factor ligand/receptor (PDGF/PDGFR) signal transduction systems are involved in a number of cellular processes such as migration, proliferation, development, and survival. PDGF ligands induce receptor dimerization, which then undergoes autophosphorylation [83]. Activated PDGFRs stimulate multiple key pathways, including PI3K/Akt, RAS/MAP kinase, Src kinase family, and PLC/PKC, which are closely implicated in tumorigenesis and normally dysregulated in cancer.

Platelet-derived growth receptor alpha (PDGFRA) mutations, amplifications, and upregulation are present in a significant subset of pHGG [84,85], with approximately 10% of mutations and 7% of amplifications in hemispheric pHGG. PDGFRA mutations were associated with worse prognosis than PDGFRA Wt. PDGFRA mutation is significantly associated with TP53 (>70%), ATRX (>60%), and H3F3A G34R/V (>40%) mutations, exhibiting association with the G34 epigenetic subtype [84]. PDGFRA amplifications show the same trend, being more associated with P53 (approximately 61%) and H3F3A G34R/V (approximately 57%) mutations than to ATRX (approximately 38%). There is also a significant overlap between PDGFRA amplifications and mutations (approximately 25%). Among the PDGFRA mutated/amplified pHGG that are Wt on H3F3A, the predominant mutations are ATRX (approximately 50%), P53 (approximately 50%), and CLTCL1, a gene that encodes a protein that constitutes the coated pits and endocytic vesicles [86]. Endocytosis dysregulation is believed to play an important role in cancer [87], but the particular role of this mutation on pHGG was not studied. The rest of the mutations co-localizing with PDGFRA mutations/amplifications are related to epigenetic mechanisms, such as KMT2D, SETD2, and HIRA; with DNA repair, such as ATM and NF1; or with cell signaling.

Constitutive PDGFRα activation blocks differentiation of glial precursors into oligodendrocytes and astrocytes [88] and is sufficient to induce glioma formation in vivo, indicating that PDGF dysregulation can be a driving mutation in pHGG [88,89]. Combined expression of PDGF-B ligand and P53 knockout further increased the penetrance of glioma formation in vivo and increased the aggressiveness of the HGG in murine models [90], indicating a cooperation between these oncogenic lesions. 

### 2.5. PTEN Mutations and the PI3K/Akt Pathway

The phosphatidylinositol 3-kinase (PI3K) pathway controls several critical cellular functions such as growth, induction of cell death, cell motility, neo-angiogenesis, and stem cell self-renewal.

PI3K is activated by growth factor receptors, including EGFR, EGFRvIII, and PDGFR and by RAS activation and catalyzes the generation of phosphatidylinositol-3,4,5-triphosphate (PIP3). Akt is activated by PIP3, initiating a signal transduction that promotes proliferation and inhibiting apoptosis.

PTEN catalyzes the dephosphorylation of PIP3, regulating the PI3K pathway, and thus when PTEN function is loss, the Akt pathway becomes constitutively active and pHGG with PTEN loss show elevated Akt levels. Additionally, PTEN was shown to translocate to the nucleus, where it plays a role in modulating centrosome stability, eliciting homologous recombination (HR) DNA repair by inducing Rad51 expression and inducing cell cycle progression [91,92]. It is believed that these functions of PTEN are independent of the PI3K catalytic function. PTEN has been associated with increased aggressiveness in HGG and poorer outcome [93]. Although PTEN mutations are more frequent in adult HGG (approximately 50%), they are found in approximately 9% of non-brainstem pHGG. Nevertheless, overactivation of the Akt pathway by other mechanisms (such as PTEN epigenetic silencing) is more frequent in pHGG and was associated with poorer prognosis [91,94].

### 2.6. BRAF-V600

The MAPK pathway is comprised of a number of signaling cascades, with Ras-Raf-Mek-extracellular signal-regulated kinase 1 and 2 (ERK1/2) being one of the most disrupted in human cancer [95]. Activating mutations in one of the key components on this pathway, the Ras GTPases were found in wide range of human cancer [96,97].

In normal conditions, Ras is activated by activation of receptor tyrosine kinases (RTK) by ligand binding. This causes Ras to switch to a GTP-bound active state. In its active conformation, Ras binds Raf and the RAS/Raf complex translocates from the cytoplasm to the cell membrane. Once in the cytoplasmic membrane, Raf acts as a kinase, activating the ERK1/ERK2 kinases, which in turn translocate to the nucleus and phosphorylate effector proteins that ultimately activate cellular responses, initiating cell programs related to cell proliferation, survival, differentiation, cell migration, and angiogenesis, among others [96,98,99]. It was also shown that Ras activates the PI3K pathway independently of the MAPK pathway [100,101,102]. Given its wide effects, Ras-activating mutations are strong oncogenes. NRAS (Neuroblastoma-Ras) activation in conjunction with P53 knockdown is enough to induce the formation of high-grade gliomas when expressed in neonatal mice neural-precursor cells on the Subventricular zone (SVZ) [14,73,103].

Downstream of Ras proteins in the MAPK pathway, Raf kinases mutations were also observed in cancer. BRAF (v-raf murine sarcoma viral oncogene homolog B1 gene/protein) Raf kinase-activating mutations occur in approximately 9% of NBS pHGGs, being the most commonly observed mutation BRAF V600E [10]. This mutation renders the BRAF protein in a constitutively active state. The kinase activity of BRAF V600E is significantly increased and activates ERK activity independently of RAS signaling. It was suggested that V600E substitution acts as a phospho-mimetic.

Besides its oncogenic effects, MAPK pathway activation may induce senescence, a mechanism modulated by cyclin-dependent kinase inhibitors, and is a mechanism that normal cells adopt to prevent abnormal oncogenic activation. On some cells, BRAF V600E mutations may induce proliferation initially, but further tumor progression is impeded by MAPK overactivity. Further oncogenic lesions such as CDKN2A p16INK4A loss are required in order to reactivate proliferation, indicating that BRAF V600E mutations require PI3K/Akt dysregulation for the tumor to progress.

### 2.7. NF1 pHGG

Neurofibromatosis type 1 (NF1) is a dominant autosomal genetic condition that affects children and adults [104]. Individuals with NF1 have an increased risk for developing central nervous system neoplasm, and although the most common brain tumors associated with NF1 are low-grade glioma, pHGG can also emerge [105]. The NF1 gene encodes neurofibromin, a protein that negatively regulates the RAS pathway through activation of GTPase-activating protein (GAP), which converts GTP-RAS to GDP-RAS, inactivating the signal cascade. Loss of NF1 leads to activation of the MEK-ERK and PI3K-Alt-mTOR pathways, and for this reason, NF1 can be considered a tumor suppressor. NF1-mutant pHGG is highly associated with P53 and ATRX inactivation. Interestingly, NF1 loss shows little association with H3F3A-G34R/V mutations.

### 2.8. Neurotrophic Tropomyosin-Related Kinase (NTRK) Fusions

NTRK genes encode proteins of the tropomyosin receptor kinase (Trk) family, composed by TrkA, B, and C receptors and encoded by the NTRK1, 2, and 3 genes, respectively. NTRK gene expression is normally silenced in neuronal lineages and in pHGG cells, and NTRK rearrangements result in fusions with genes actively transcribed in pHGG cells, such as neurofascin (NFASC) and brevican (BCAN) for NTRK1 [106]. This results in the activation of NTRK expression. The fusion also causes loss of a polypeptide tract of NTRK receptors, rendering them in a constitutively active conformation. TRK pathway activation operates on downstream signaling pathways, markedly activating the Ras pathway, which also results in ERK and AKT activation. Thus, NTRK fusions are related to other molecular alterations on pHGG which also alter these pathways. Approximately 10% of non-brainstem pHGG carry out NTRK fusions. Notably, this percentage increases to 40% for non-brainstem pHGG on infants younger than 3 years old [52,107]. Tkr rearrangements play important roles in oncogenesis in various types of tumors, including pHGG [106]. Ectopic expression of NTRK fusions was shown to increase proliferation and colony formation in vitro as well as tumor formation in vivo [106], a phenotype similar to the one resulting from Ras constitutive activation. Entrectinib, an NTRK inhibitor, was recently approved by the U.S. Food and Drug Administration (FDA) for treatment of NTRK fusion-positive tumors [108,109]. 

### 2.9. IDH Mutations

The isocitrate dehydrogenase genes, IDH1 and IDH2, are mutated with high frequency in adult secondary HGG. These tumors arise normally lower-grade glioma and recur as HGG. The IDH mutation is present in all the lineages derived, indicating that IDH acts as a cancer-driving mutation. Mutations on IDH enzymes result in acquisition of a new function: mutated IDH drive the conversion of α-ketoglutarate to r (-)-2-hydroxyglutarate (2-HG), an oncometabolite absent in normal conditions. 2-HG accumulation causes profound epigenetic changes mediated by inhibition of DNA and histone methylases. This impacts cell differentiation, development, proliferation, and DNA repair response and immune-response programs. IDH mutations are rare among cortical pHGG (<4%), and a study of the molecular basis of IDH driven tumors was done mainly in the context of adult LGG and recurrent HGG. As happens in the case of adult HGG, cortical pHGG harboring IHD mutations exhibits better survival compared with IDH Wt pHGG but worse than the subtypes associated with NF1 and BRAF mutations, which belong to the subtypes LGG-like and PXA-like. IDH mutant-related mechanisms were studied in detail in the context of adult glioma. For this reason, a discussion of these mechanisms in this article is beyond our aim.

### 2.10. H3F3A K27M Mutations

K27M mutations in H3.3 and H3.1 are predominant in brainstem pHGG. H3F3A-K27M-mutant cortical pHGGs are rare (approximately 1.3% incidence). Lysine 27 is an amino acid that is commonly methylated and acetylated. The first mark is associated with transcriptional repression, and the second is associated with active enhancers. Predictably, these epigenetic signals are disrupted in K27M pHGG [110,111]. This has consequences on cell differentiation, proliferation, and neoplastic activation, i.e., K27M increased the frequency of transformation of NPC into a malignant phenotype. Significant efforts have been directed toward identifying the epigenetic mechanisms underlying H3K27M mutations and their roled in gliomagenesis and pHGG development in the context of midline pHGG, which led to great discoveries over the last few years. For this reason, we will not go into detail on the state-of-the-art of this mutation in this work. The reader is referred to the following articles for further details [112,113,114].

## 3. Origins of Cortical pHGG

Gliomas are broadly histologically classified based on characteristic similarities to glial cells and are more specifically classified as astrocytic, oligodendroglial, or a mixed oligodendroglial-astrocytic type, referring to the lineages within the CNS [2,115]. Neural stem cells (NSCs) or oligodendrocyte precursor cells (OPCs) have been posited as the most likely candidates to be the cellular origin of glioma [116,117]. A number of properties of NSCs has historically rendered them the putative cell of origin, including their self-renewal potential, plasticity, shared molecular characteristics with human cancer stem cells, as well as the ability to produce gliomas when transfected with oncogenic DNA and transplanted into adult mice [118]. However, recent findings have challenged the argument for NSCs to increasingly support OPCs as the cell of origin in pediatric glioma. Unlike NSCs, OPCs retain a significant proliferative population in the postnatal brain in humans and mice [119,120,121]. Furthermore, there is now evidence suggesting that OPCs demonstrate some lineage plasticity, are susceptible to molecular reprogramming to exhibit more NSC-like properties, and can give rise to glioma subtypes besides oligodendroglioma and proneural high-grade glioma [122,123,124,125].

While the cellular origins of pHGGs are still relatively unknown, the observation that pHGGs arising in different areas of the brain have distinct molecular and clinicopathological features has suggested the unique developmental origins of pHGG subtypes (Figure 3) [117,126]. Notably, spatial-temporal gene analysis of H3F3A mutant pHGG revealed distinct developmental expression profiles for G34R and K27M tumors (Figure 3). H3F3A G34R mutant tumors correlated to expression patterns of early embryonic and fetal development of the neocortex and striatum [54] as well as early fetal development of the amygdala, inferior temporal cortex, and the ganglionic eminences [18,56]. Meanwhile, expression signatures of H3F3A K27M mutant tumors were associated with embryonic development of the upper rhombic lip [56] as well as fetal development of the thalamus, striatum, and cerebellum [54,56]. The relationship between anatomical location of these molecular subtypes of pHGG emphasizes the importance in understanding the cell of origin of pHGG and the distinct developmental programs exploited in malignant transformation. Determining the cell of origin and understanding the neurodevelopmental context of pHGGs may provide critical insight into the heterogeneity of these tumors and may unveil novel therapeutic targets [6].

Interestingly, G34R-mutant tumors have been found to express a number of genes for developmentally regulated transcription factors and surface markers, suggesting that these tumors exist in an NSC-like state. Combined methylation and gene expression profiling identified differential DNA hypermethylation at the loci for Oligodendrocyte Lineage Genes 1 and 2 (OLIG1 and OLIG2) and corresponding decreases in expression in G34R mutant tumors [54]. This epigenetic repression of OLIG1/OLIG2 genes has been proposed as a mechanism by which G34R mutant cells avoid lineage commitment and retain pluripotency [54,127]. Furthermore, differential binding of H3K36me3 has also been found to drive abnormal expression of stem-related genes in G34R mutant cells, including Musashi-1 (MSI1), eyes absent homolog 4 (EYA4), and, notably, SRY-Box Transcription Factor 2 (SOX2) required for stem cell maintenance and marker of NSCs [56,128]. Additionally, chromatin immunoprecipitation sequencing (ChIP-seq) analysis of differential H3K36me3 binding in G34R mutant patient samples demonstrated elevated expression of a number of transcription factors related to forebrain development, including FOXG1, DLX6, ARX, DLX5, FOXA1, NRSE1, POU3F2 and SP8, and MYCN [56]. However, using a genetically engineered mouse model (GEMM), it was demonstrated that the introduction of G34R mutation and p53 loss in NPCs in the embryonic forebrain was unable to produce tumors [114,129]. Collectively, such findings have led researchers to consider that the NSC-like features of H3 G34R-mutant pHGG are unlikely to be inherited from the cell of origin but may be the result of epigenetic reprogramming, mediating the dedifferentiation of OPC cells [128].

## 4. Tumor Heterogeneity Uncovered by Single Cell Analysis

Over the last years, it has become increasingly evident that high-grade gliomas exhibit different levels of intratumoral heterogeneity. From a genetic perspective, various clonal populations are observed in single HGG tumors, reflecting mutational events that have occurred over HGG progression. Another layer of heterogeneity was evidenced on the activation states of receptor tyrosine kinases (RTK) [130]. More recently, tumor heterogeneity was characterized by single-cell RNA sequencing (scRNA-seq) [131]. In this study, both pediatric and adult HGG were subjected to scRNA-seq and all tumors exhibited 4 different expression signature clusters, regardless of the HGG subtype (i.e., 4 main different cell types with different transcriptional landscapes). These transcriptional states resemble OPC, neural progenitor cell, astrocytic, or mesenchymal lineages, respectively. Each of the tumors analyzed contained every cell type except for some pediatric tumors lacking astrocytic-like cells. Interestingly, cells in between these states were found in most tumors, indicating that cells within a transcriptional state can transition to another. To confirm this, cells belonging to each transcriptional cluster were isolated, and it was demonstrated that each one of the clusters has the ability to regenerate the other three. This implies that tumor-initiating cells (or cancer stem cells) are present in every transcriptional state and that a homeostasis exists where the four main transcriptional states cooccur. Notably, a small number of oncogenic lesions, namely PDGFR, CDK4, EGFR, and NF1, were demonstrated to determine the proportions of the four main transcriptional states [131]. In this sense, PDGFRa-amplified/mutated tumors have OPC-like cells predominantly, CDK4 amplified tumors are constituted mainly of NPC cells, EGFR-amplified tumors are mainly astrocytic-like cells, and NF1 is mainly mesenchymal cells (Figure 3). Nevertheless, plasticity of the cells to transition between states makes it possible for a tumor to alter these proportions in response to treatments as a mechanism of escape. 

## 5. Molecular NBS pHGG Targets

The molecular landscape of NBS pHGG is heterogeneous, and therapies must be accordingly diverse and specific. A corollary of this is that the more refined the molecular diagnosis of the particular NBS pHGG, the better the chances of designing a therapy that would target the patient’s pHGG molecular alterations.

Highly targetable molecular alterations are found in different subtypes of NBS pHGG. For these tumors, identification of the alterations is a very determinant factor of patient outcome.

As mentioned before, BRAF-activating mutations, such as BRAF V600E, are common in LGG and present in NBS pHGG. BRAF inhibitors can block BRAF-mediated activation of MAPK/ERK pathway, and these compounds showed good clinical responses. It must be observed that BRAF inhibitors not only have poor activity against BRAF-WT cells but also result in activation of the Raf-MEK-ERK pathway, enhancing tumor proliferation and aggressiveness. For this reason, molecular diagnosis of the BRAF status is imperative for application of this tailored therapy. Clinical trials are currently recruiting to assess the BRAF inhibitor monotherapy on BRAFV600E mutant pHGG [108].

Homozygous loss of NF1, which also results in RAS constitutive activation and MEK/ERK upregulation, can be targeted with MEK inhibitors. This treatment showed efficacy on LGG, and targeting NF1 NBS pHGG resulted in tumor regression [132]. NF1 tumors also exhibit increased immunological activity [133], which makes them more suitable for immune therapies.

Evidence indicates that histone mutations occurring in pHGG play their oncogenic roles through altering predominantly their own and other histones marks. This not only causes aberrant epigenetic silencing or activation of genes but also may affect chromatin structure. Thus, strategies to target histone-mutant NBS pHGG should point towards reverting the gene expression alterations caused by these mutations on key genes and/or molecularly neutralizing the dominant effects caused by these mutations [134]. Histone deacetylases (HDAC) are responsible for removing acetyl marks from histone tails and HDAC inhibitors have been shown to restore these marks in K27M mutant cells in vitro [135]. Clinical trials assessing the effect of HDAC inhibitors in DIPG treatment are ongoing, although the benefit of these compounds is not clear. On the other hand, NBS pHGG harboring H3F3A G34R/V mutations do not demonstrate benefit from these drugs [134]. More molecular studies are required to shed light on the effects of HDAC inhibitors on histone-mutant pHGG. Particularly, the ability of HDAC inhibitors to revert the cell to a histone wild type-like status and the extent of off-target epigenetic changes resulting from treatment is still a matter of debate [134]. Other drugs may emerge to reverse epigenetic alterations caused by histone mutations or strategies to block the interaction of the mutant histone with target proteins. For now, it seems that operating on the genes that are dysregulated in histone-mutant cells provides a more obtainable way to target these cortical pHGGs. For example, H3F3A G34V mutation has been reported to epigenetically upregulate MYCN expression, and this mechanism results in increased cell proliferation [56]. MYCN has been targeted by inhibiting proteins that are associated with MYCN stabilization, such as checkpoint kinase 1 (Chk1) and AURKA kinases. An AURKA inhibitor, VX-689, demonstrated selectivity against pHGG cells expressing H3.3-G34V. These drugs can be also be useful in treating MYCN-amplified NBS pHGG [56].

It must be noted that the genomic alterations and gene expression profiles can be informative but not determinant in predicting sensitivity to a particular treatment. Patients with NBS pHGG carrying similar mutations may respond differently to precision therapies, the current best-case scenario being only a subset of patients within a molecular subgroup responding to the specific therapy. Each patient has a unique combination of genetic and epigenetic alterations, and the extent of response to a particular treatment will be a consequence of this complex landscape. Other factors that make it difficult to predict particular responses are tumor intraheterogeneity and the patient’s own distinct genetic and environmental background. For these reasons, patient-tailored treatments have a strong likelihood of providing better clinical responses. Currently, patient-derived tumor cells can be grown ex vivo or in vitro to assess the individual’s response to different treatments. These platforms can also be used to track tumor evolution driven by the response to therapies. The development of resistance to treatments is a common feature of recurrent gliomas. Staying informed on whether a treatment has loss efficiency can overcome this resistance side-effect and dictate the use of more effective therapies. One of the disadvantages of this approach is that the fast NBS pHGG progression limits the time available to gather information from the experimental systems. Testing treatments in animals can take more than 100 days, and advanced pHGGs may not gain benefit from the findings. In vitro systems are more limited, but high-throughput drug screenings can be completed in less than two weeks. Cancer precision medicine platforms are expanding quickly, and new technologies will emerge to generate relevant decision-guiding information in less time, and their use will expand to more patients.

## 6. Discussion

Our knowledge of cortical pHGG has drastically increased over the last decade. A major breakthrough was the identification of predominant genetic alterations and epigenetic signatures on cortical pHGG and the concomitant refinement of their molecular classification [10,21]. The identification of epigenetic mechanisms disrupted in cortical pHGG opened up a range of possibilities for therapeutic interventions. Although part of the molecular mechanisms linking epigenetic alterations with oncogenesis in these tumors was previously described [10,54], we are still far from having a clear picture of the role of epigenetic alterations on cortical pHGG development. The study of the effects of epigenetic alterations found in cortical pHGG on specific transcriptional programs and their dependency on the cellular contexts and developmental stages will provide insights on the effects of these alterations in triggering and maintaining pHGG. The structural effects on chromatin that these alterations cause may also play a role in oncogenesis and tumor progression, and these effects may as well be cell context-dependent. Additionally, alterations on chromatin structure may have consequences on the ability of the cell to repair DNA and may lead to genomic instability. Deficiencies in DNA repair and genomic instability not only can be targeted with DNA repair and cell cycle checkpoint inhibitors but also can be exploited to prime immune system responses [51,136]. 

In order to unveil the detailed mechanisms linking epigenetic alterations and pHGG oncogenesis, the development of refined models is imperative. Studying the effect of a genetic alteration in cells extracted from an already developed cortical pHGG may not reveal the role that this alteration played in driving the tumor at early stages or cause the same epigenetic effects that it causes in the normal context, where the transformed cell is a naïve precursor cell. For this reason, models that recapitulate natural neoplastic transformation may be more informative on the role of mutations in the context of early tumor development.

Although modelling cortical pHGG will provide more detail in relation to the molecular mechanisms underlying these cancers, classification of the molecular alterations already available allows precision therapy interventions in some cases. For this reason, high-throughput molecular profiling of cortical pHGG patients is closer to becoming a standard practice [137]. When possible, this can be accompanied by in vitro cell culture of biopsied samples and assessment of the cell’s therapeutic vulnerabilities. Although this type of patient-tailored approach is at the early stages, the technological advances and improvements on cost efficiency of omics platforms and the implementation of integrative medicine centers with high capacity on molecular biology equipment will make these practices more accessible.

### Future Perspectives

Intratumoral heterogeneity in pHGG is a key concept that will likely change the way pHGG treatment is conceived and implemented. As recurrence and development of resistance are major mechanisms accounting for treatment failure, it can be expected that identification and targeting of specific cellular populations within the tumor that develop resistance mechanisms and sustain tumor proliferation will provide a novel way to prevent recurrence. Additionally, stem cells within the pHGG, which have the ability to repopulate the tumor cellular subtypes, are strong candidates for targeted therapies. Although the concept of analyzing heterogeneity within pHGG to implement treatment decisions appears to be unavoidable [138], the technologies to unveil single cell heterogeneity, such as single cell RNA-seq, are not frequently available in clinical facilities, and analysis of the information it provides is still far from being straightforward.

Additionally, two research areas are essential for the future of cortical pHGG treatment. One of them is drug delivery to the CNS. Novel strategies aiming at bypassing the blood–brain barrier (BBB) are under intense study, and nano-technologies will likely be implemented in the near future to control intratumoral drug administration. Gene therapy approaches also had major improvements over the last years, and in the future, reversing the effects of genetic alterations observed in cortical pHGG will become feasible. In particular, genetic/epigenetic therapy has experienced a significant breakthrough within the last years, i.e., the CRISPR-Cas technology can be used not only for gene edition but also to change the epigenetic status of a particular gene [139]. The development of systems to deliver CRISPR-Cas constructs into pHGG will be critical for the success of epigenetic and genetic therapies.

The second field that will likely be determinant in improving cortical pHGG outcome will be the study of the immune system in these tumors. The molecular mechanisms underlying immune system tumor-mediated suppression and the molecular intervention based on stimulating antitumor immunity has proven successful in various solid tumors, but the particular immune context found in the CNS imposes additional challenges for cortical pHGG. Single cell sequencing has the ability to provide a detailed depiction of immune cell populations and the overall immune antitumor activity, and its use might become critical to decide on strategic immune therapeutic interventions [138].

## Figures and Tables

**Figure 1 ijms-21-09654-f001:**
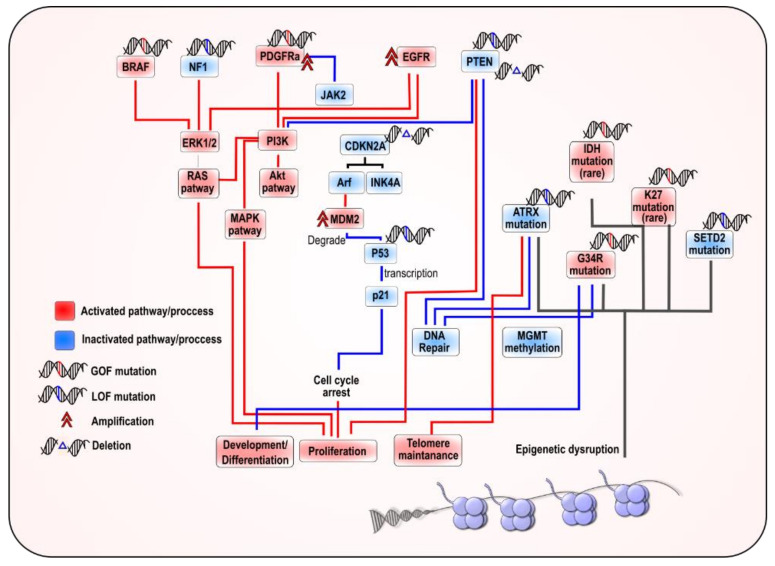
Illustration depicting the main pathways altered in hemispheric pediatric high-grade gliomas (pHGG): The main genetic alteration associated with pathways alterations are indicated.

**Figure 2 ijms-21-09654-f002:**
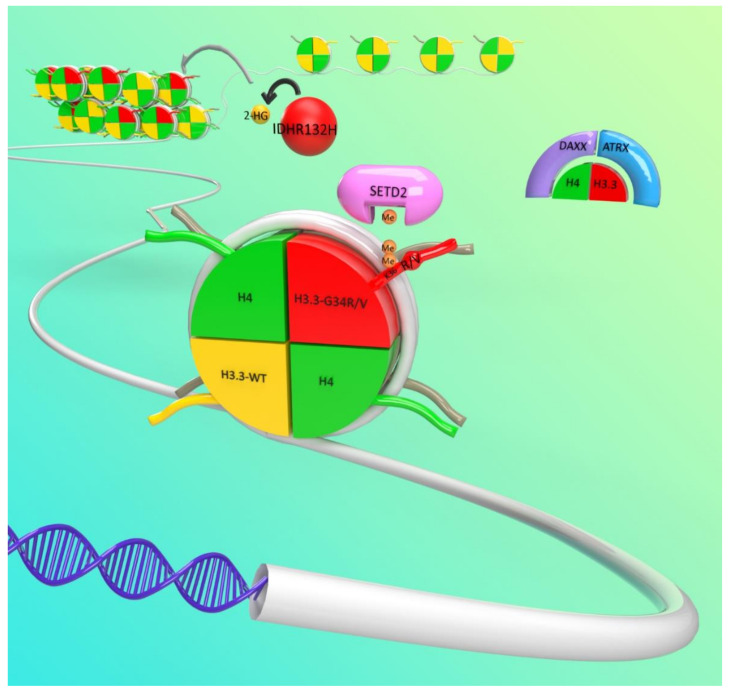
Illustration depicting the most prominent epigenetic alterations in hemispheric pediatric high grade glioma (pHGG): The most common epigenetic alterations are H3F3A-G34R/V (Glycine to Arginine or Valine mutations in Histone H3.3) mutations and ATRX (thalassemia/mental retardation syndrome X-linked) loss-of-function mutations. Less frequently, mutations on SET Domain Containing 2 protein (SETD2), an H3K36me3 writer, Death domain-associated protein (DAXX), part of the ATRX/DAXX complex, and Isocitrate dehydrogenase 1 (IDH1), among others, also act as epigenetic disruptors. IDH1 gain of function mutations lead to the production of 2-hydroxyglutarate (2-HG), which inhibits histone and DNA demethylases.

**Figure 3 ijms-21-09654-f003:**
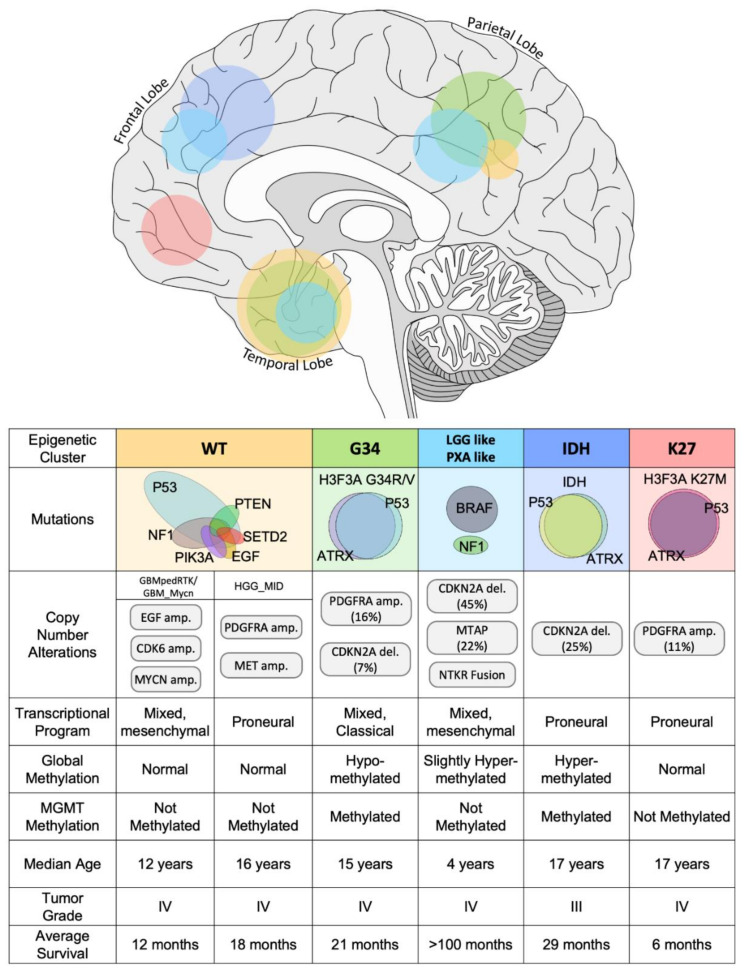
Scheme depicting the molecular classification of hemispherical pHGG, indicating the main molecular subgroups and the common mutations, genetic alterations, transcriptional programs, global, and O-6-Methylguanine-DNA Methyltransferase (MGMT) methylation status, and the clinical characteristics (included anatomical location in the upper brain illustration) associated to each molecular subgroup.

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
