# Peer review of "Hemispherical Pediatric High-Grade Glioma: Molecular Basis and Therapeutic Opportunities"

_ijms, 2020, doi:10.3390/ijms21249654_

Round 1
Reviewer 1 Report
Authors of the manuscript present a comprehensive and accurate review of most of the clinical and molecular findings in children with HGG, focusing on the molecular aspects and giving accurate and updated information
I have no specific remarks a part from the excessive length of the manuscript.
Author Response
We have addressed your comment regarding the length of the manuscript, by removing section 5, (pHGG models) which we consider is not essential for the review.
Reviewer 2 Report
This review provides a very good and structured overview of pediatric high grade gliomas. I think that the authors should do more justice to the title passage of the "therapeutic opportunities" by adding a sub-header "Future perspective" or similar to the last section of the very short discussion and present the listed two points in more detail in a larger context. The important information may get lost in the presented amount and so the interested reader will get the essence of the new information (again) in the last section.
Author Response
We acknowledge the reviewer for the positive comments,
The reviewer states that we should do more justice to the title passage by adding a sub-header "Future perspective" or similar to the discussion section of the very short discussion and present the listed two points in more detail in a larger context. The important information may get lost in the presented amount and so the interested reader will get the essence of the new information (again) in the last section.
We improved the discussion section as requested, by adding the subsection "Future perspectives" into the discussion section, and we extended both sections. We also addressed the issue regarding the length of the manuscript, by removing section 5 (pHGG models), which we consider is not essential for the review.